# Automatic Patient-level Diagnosis of Prostate Disease with Fused 3D MRI and Tabular Clinical Data

**Oleksii Bashkanov**[*1]                OLEKSII.BASHKANOV@OVGU.DE

**Marko Rak**[1]                     RAK@ISG.CS.OVGU.DE

**Lucas Engelage**[2]                LUCAS.ENGELAGE@ALTAKLINIK.DE

**Christian Hansen**[1]               CHRISTIAN.HANSEN@OVGU.DE

[1] *Faculty of Computer Science and Research Campus STIMULATE, University of Magdeburg, 39106, Germany*

[2] *ALTA Klinik, Bielefeld, 33602, Germany*

**Editors:** Accepted for publication at MIDL 2023

## Abstract

Computer-aided diagnosis systems for automatic prostate cancer diagnosis can provide radiologists with decision support during image reading. However, in this case, patient-relevant information often remains unexploited due to the greater focus on the image recognition side, with various imaging devices and modalities, while omitting other potentially valuable clinical data. Therefore, our work investigates the performance of recent methods for the fusion of rich image data and heterogeneous tabular data. Those data may include patient demographics as well as laboratory data, e.g., prostate-specific antigen (PSA). Experiments on the large dataset (3800 subjects) indicated that when using the fusion method with demographic data in clinically significant prostate cancer (csPCa) detection tasks, the mean area under the receiver operating characteristic curve (ROC AUC) has improved significantly from 0.736 to 0.765. We also observed that the naïve concatenation performs similarly or even better than the state-of-the-art fusion modules. We also achieved better prediction quality in grading prostate disease by including more samples from longitudinal PSA profiles in the tabular feature set. Thus, by including the three last PSA samples per patient, the best-performing model has reached AUC of 0.794 and a quadratic weighted kappa score (QWK) of 0.464, which constitutes a significant improvement compared with the image-only method, with ROC AUC of 0.736 and QWK of 0.342.

**Keywords:** Computer-assisted diagnosis, prostate cancer, disease prediction, convolutional neural networks, tabular clinical data.

## 1. Introduction

Prostate cancer (PCa) is the second most diagnosed cancer among men in more than half of the world's countries, with an incidence rate approximately three times higher in transitioned countries than in transitioning countries (Sung et al., 2021). Despite the widespread nature of PCa, the understanding of the etiology of prostate cancer remains largely unknown, aside from factors such as aging, family history, and some genetic mutations (Sung et al., 2021). By studying machine learning methods to combine data from multiple sources,

---

[*] Corresponding author

we could not only increase diagnostic predictive performance but also potentially approach a better understanding of the etiology of prostate diseases.

Prostate-specific antigen (PSA) test is a common cancer screening method due to strong evidence that higher PSA levels are associated with a more advanced PCa status (higher Gleason Score) (Izumi et al., 2015). However, its levels can be elevated not only due to prostate cancer, but also due to related conditions such as chronic prostatitis or benign prostatic hyperplasia (Cabarkapa et al., 2016). As it often causes unnecessary biopsies and leads to overdiagnosis and subsequent overtreatment, the PSA marker should be treated carefully and should be considered alongside other markers such as patient demographics. It is desirable that patients have multiple PSA samples; however, they are not straight-forward to summarize (e.g., via PSA kinetics) because they act as irregular time series. Despite the clear evidence that PSA kinetics are crucial to understanding the prognosis of advanced prostate cancer, according to (Vickers and Brewster, 2012), PSA changes over time provide little to no value in the diagnosis of prostate cancer.

Alongside other clinical parameters, multiparametric magnetic resonance (MR) imaging continues to gain wider clinical acceptance in the PCa diagnostic routine, as it can greatly improve cancer detection quality. MR screening enables the planning of targeted treatment or biopsies; in some cases, it may allow biopsies to be avoided altogether because of the higher PCa detection specificity than the PSA screening method.

Deep learning is well suited for multimodal learning problems due to their flexibility. This means that, in conjunction with multiple MR modalities, tabular data could also be represented as input (Cui et al., 2022). Here, the main challenge lies in the architecture of convolutional neural networks (CNN), as those contain significantly higher and denser para-metric capacity than needed for non-image data. The intuitive way to integrate both is to concatenate image-based features outputted by the CNN with tabular features directly. After that, fully connected layers or other downstream classifier can be applied (Esmaeilzadeh et al., 2018; Mobadersany et al., 2018; El-Sappagh et al., 2020; Mehta et al., 2021).

A different, more promising research direction in this context is attention-based fusion methods, where image and tabublar data can interact more interconnectedly at the archi-tectural level (Cui et al., 2022). Duanmu et al. (2020), for instance, proposed to upscale tabular features with a fully connected layer to match the number of post-downscaling CNN features, which allows one to perform channel-wise multiplication. This approach outper-formed the pure image-based model as well as simple concatenations with tabular data. Feature-wise Linear Modulation (FiLM) was introduced by (Perez et al., 2018) to dynam-ically scale and shift the CNN features conditioned by non-image information. It was also adopted for disentangled representation learning and for segmentation in the medical do-main (Chartsias et al., 2019; Jacenków et al., 2020; Lemay et al., 2021). Given that idea, the dynamic affine transform (DAFT) module was proposed (Pölsterl et al., 2021; Wolf et al., 2022). The main feature of the DAFT module is that it adapts the scaling and shifting parameters depending on *both* image-based and tabular features. The authors hypothesize that DAFT provides a two-way information exchange between two modalities, whereas in FiLM only the tabular modality informs the image modality one-directionally.

There is little research on multimodal information fusion in the prostate diagnosis do-main. In Mehta et al. (2021)'s work, the fusion is done on the last feature level by con-catenating individually produced feature vectors by each imaging modality and tabular

information. Later, feature selection in the resulting vector and multiple levels of support vector machines were applied to predict the probability of csPCa. However, in this case, neither multimodal CNNs nor tabular data inform each other.

In this work, we explore previously named state-of-the-art methods for patient-level prostate disease classification with multiparametric MRI and tabular data. Our work is not limited to binary classification of clinically significant PCa; we also predict prostate disease by means of multiclass classification to give a broader picture of performance. Furthermore, we discuss the case of variable-length PSA data and its impact on performance, including the cases where PSA data is missing.

## 2. Methods

### 2.1. Neural network

As we operate at the patient's level, it is rational to consider 3D neural network architectures because they can capture the entire context of the prostate gland in contrast to 2D/2.5D architectures, where a single or couple of neighboring slices are given. 3D MR images of the prostate are usually highly anisotropic; our architecture must account for that. We opt for a lightweight version of ResNet (He et al., 2016). The configuration of the kernel and stride size in residual blocks was adjusted according to the encoder of the self-configurable segmentation network nnU-Net (Isensee et al., 2021). Due to the amount and nature of the data, we reduced the capacity of the model and started with four convolution filters and doubled them in the next residual block, resulting in 55,923 learnable parameters in total.

We augmented this CNN backbone with discussed state-of-the-art modules. Namely, with plain concatenation and linear / non-linear classification on the tabular input side, by an upscale feature module as in Duanmu et al. (2020), by linear modulation modules FiLM (Perez et al., 2018) and DAFT (Pölsterl et al., 2021), as can be seen in Figure 1. While most state-of-the-art approaches introduced tabular features along the CNN backbone densely, in DAFT these features were introduced only ones, before the last residual block. Pölsterl et al. (2021) argued that, in the early CNN layers, the features are rather primitive and thus do not match the information concepts of the tabular data. On the other hand, one could argue that early dense fusions leverage interconnection between image-based and tabular features. We closed this knowledge gap experimentally, comparing the performance of the late-fusion and dense-fusion (at every residual block) of the FiLM and DAFT modules. In terms of a local position on the fusion modules in the residual blocks, the FiLM and DAFT empirically favored placement before the first convolution.

### 2.2. Dataset

**Image data**  Our data set includes 3800 multiparametric MR studies of biopsy-naïve patients from ALTA Klinik (Bielefeld, Germany). All scans were obtained according to the recommendations of PI-RADS v2. All patients gave their informed consent to use their images for research purposes. The anisotropic spacing for the T2-weighted sequences is around $0.5 \times 0.5 \times 3.2\,mm^3$ and $1.0 \times 1.0 \times 3.2\,mm^3$ for the apparent diffusion coefficient (ADC). Both image modalities were resampled to $0.5 \times 0.5 \times 3.0\,mm^3$ and cropped to form a central $140 \times 140 \times 20$ tensor. T2-weighted images were normalized by the Z-score per

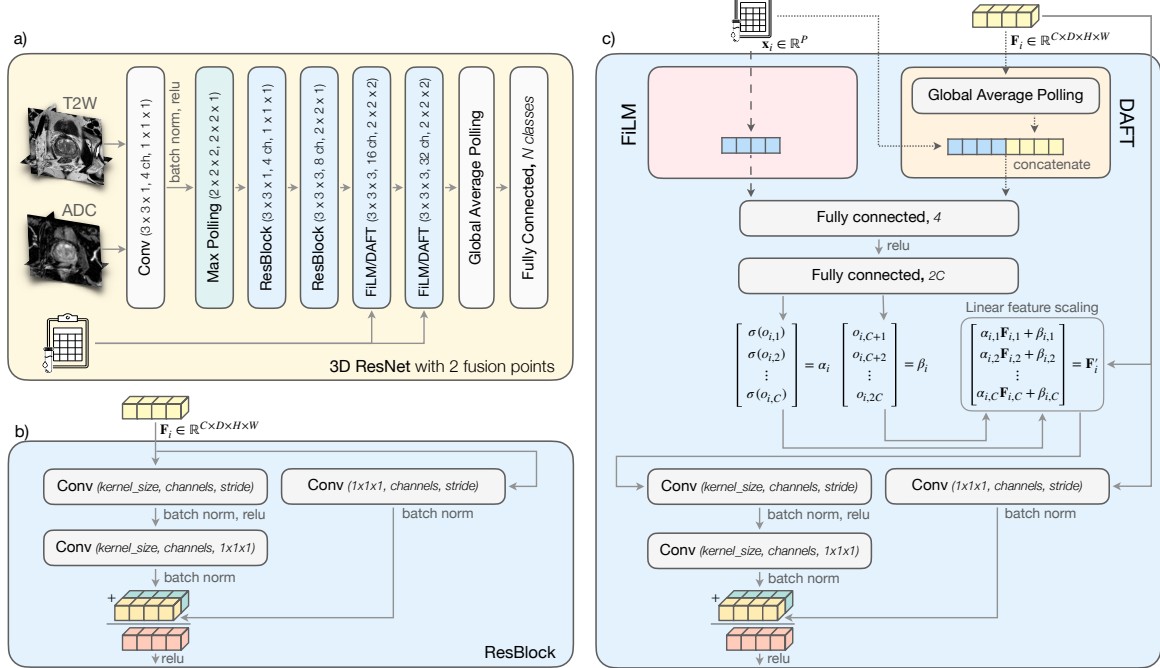

Figure 1: Main elements of the examined architectures: a) Adopted 3D ResNet-like back-bone with 2 fusion points; b) Residual block; c) Fusion block with linear feature modulation (FiLM/DAFT). Dashed lines with low and high frequency depict the mutually exclusive flow of the data for FiLM and DAFT blocks, respectively.

sample, while the intensity of the ADC was clipped at a value of 3000 and normalized to the [0; 1] range to preserve its quantitative properties. Random cropping and horizontal flipping with probability of 0.5 were used to augment the data spacially during training. As for intensity-based augmentation we used Gaussian smoothing with anisotropic filter of $([0.1, 2], [0.1, 2], [0.1, 2/6])$ and random Gaussian noise with $\sigma$ of 0.1 and zero mean.

**Tabular data** Our tabular data consist of year of birth, age, weight, height, and body mass index of the patient. For each feature, we introduced a binary one that indicates whether a value is missing. Thus, we have 10 features in total. For missing values, we used features sample mean w.r.t. the train split, which agrees with Jarrett et al. (2019) and Pölsterl et al. (2021). All non-binary features were treated as numerical and were normalized with the Z-score of non-missing samples from the current training split. During experiments, we varied the length of the PSA history, covering the latest one, two or three PSA values before MRI acquisition. Table 5 in Appendix B provides an overview of tabular data.

**Clinical annotations** Before biopsy, prostate MRIs were evaluated according to PI-RADS scores to reflect the reader's interpretation of the probability that csPCa is present. The aggressiveness of the prostate cancer was retrieved from the results of the histopatho-logical examination, which is well-suited as ground truth for predictive modeling. The aggressiveness was expressed through the Gleason Score (GS), which covers the two scores of the most prevalent cancerous tissue patterns. The following scores are possible: $GS6$, $GS7a$, $GS7b$, $GS8$, $GS9$, and $GS10$. Gleason scores of $GS7a$ and above mark clinically

significant cancer. Biopsy examinations can also indicate a related condition, *chronic prostatitis*, and thus express even finer granularity in low-risk cases. Therefore, we included prostatitis as an additional target variable. Moreover, it is one of the common triggers for false positive csPCa findings during the interpretation of MRI (Epstein et al., 2016). On the aggressiveness scale, we defined the prostatitis as less severe than $GS6$. As we operated at the patient level and not at the instance level for the related MR study, the highest-grade tumour (index lesion) for each subject was identified. A class distribution breakdown can be found in Table 4 in Appendix B.

## 2.3. Experimental setup

We applied a five-fold cross-validation scheme with hold-out test set of 640 cases. For each fold, it resulted in approximately 2,520 training cases and 640 validation cases. We stratified all the splits by target prostate diseases to ensure a similar distribution for each class. We also made sure that the data from the same patient remain in the same split. The reported performance is averaged across five folds using a hold-out test set.

In our experiments, two tasks were considered: 1) *binary* prediction of the probability associated with the presence of csPCa in the prostate gland and 2) full-grain *multiclass* prediction of prostate diseases, including prostatitis and GS classification. For these tasks, binary cross-entropy and categorical cross-entropy were optimized using Adam with decoupled weight decay (AdamW) (Loshchilov and Hutter, 2018) with learning rate of $5.5 \times 10^{-3}$ and weight decay $1 \times 10^{-4}$. We trained our models with batch size of 256 for 80 epochs. The best model was selected based on validation task-specific metrics: area under curve of receiver operating characteristic (ROC AUC) for prediction of csPCa and the combination of ROC AUC and the quadratic weighted kappa score (QWK) for prostate diseases grading. We seeded all random operations (data augmentation, batch sampler, optimizer) to ensure an adequate experimental control.

## 2.4. Evaluation

For evaluation, we reported precision (P), recall (R), ROC AUC, PR AUC, accuracy (ACC) and QWK where appropriate for the binary and multiclass predictions. To make the results of binary and multiclass tasks results comparable, we furthermore mapped the multiclass task into binary csPCa classification by summing the probabilities of Gleason scores of $GS7a$ and above.

## 3. Fusion methods

**Experiments** We compare DAFT and FiLM with the baseline and unimodal approaches. For models with tabular features only, we used logistic regression as classifier, while for models with image-based features only, we used our ResNet. For the Concat-1FC model, tabular features are concatenated with the image features before the final classification layer. In 1FC-Concat-1FC, tabular features are fed into a fully connected layer with non-linear ReLU activation. Then, similar to Concat-1FC, we concatenate processed features with the image features. In 1FC-Concat-1FC, FiLM and DAFT tabular features go through non-linear transformations with the bottleneck size of 4. Linear modulation in FiLM and

DAFT is applied directly without activation. We also include the variant of Duanmu et al. (2020).

To test the impact of dense fusion of tabular features for DAFT and FiLM, we introduce the FiLM Dense and DAFT Dense as dense variants of the FiLM and DAFT methods. As ResNet consists of 4 residual blocks, we can define up to 4 fusion points (FPs). This allows us to test how early in CNN the features should be modulated by tabular data. Each of the dense-fusion methods has 3 variants with 4, 3, 2 fusion points at 4, 3, and 2 last residual blocks accordingly. For the sake of visualization, Figure 1 shows 3D ResNet with 2 fusion points in the last two residual blocks. It should be noted that FiLM/DAFT Dense with 4 FPs refers to the original approach (Perez et al., 2018), while the simple FiLM/DAFT notation refers to the fusion in one place as in (Pölsterl et al., 2021).

**Results** The results for multiclass and binary tasks are presented in Table 1 and Table 3 in Appendix 3, respectively. For both tasks, we observed a clear performance boost when introducing demographic profiles of patients into the Concat-1FC or 1FC-Concat-1FC models. The DAFT method almost reached the performance of the best performing method 1FC-Concat-1FC on both tasks. However, in relation to the FiLM it performed only slightly better. Unexpectedly, most dense-fusion methods failed to outperform image-only methods, except DAFT with fusion at the last two residual blocks. Table 1 shows no considerable difference between FiLM Dense FPs-4, DAFT Dense FPs-4, and logistic regression in tabular data. The ablation results on dense fusion with different FPs provided a clear trend in favor of a late single fusion point on both tasks. These findings may indicate that too narrow interconnectedness between image-based and tabular features may actually undermine the overall CNNs performance, specifically when tabular data is only weakly correlated

Table 1: Performance for multiclass tasks averaged accross five folds on hold-out test set. Column T marks the linear (L) / non-linear (NL) transformation of tabular data. Column FPs shows the number of fusion points, where FP > 1 means dense fusion.

| | | | csPCa AUC | | | | | |
|---|---|---|---|---|---|---|---|---|
| **Multiclass** | T | FPs | ROC | PR | Precision | Recall | ACC | QWK |
| Logistic regression | L | - | 0.651 | 0.505 | 0.607 | 0.584 | 0.636 | 0.227 |
| ResNet | - | - | 0.736 | 0.611 | 0.679 | 0.684 | 0.685 | 0.342 |
| Concat-1FC | L | 1 | 0.767 | 0.644 | 0.701 | **0.706** | 0.695 | 0.376 |
| 1FC-Concat-1FC | NL | 1 | **0.773** | **0.711** | **0.706** | 0.665 | **0.721** | **0.419** |
| Duanmu et al. (2020) | NL | 4 | 0.720 | 0.601 | 0.659 | 0.654 | 0.677 | 0.312 |
| FiLM Dense | NL | 4 | 0.627 | 0.473 | 0.595 | 0.591 | 0.621 | 0.235 |
| FiLM Dense | NL | 3 | 0.713 | 0.580 | 0.65 | 0.649 | 0.667 | 0.302 |
| FiLM Dense | NL | 2 | 0.730 | 0.603 | 0.664 | 0.661 | 0.682 | 0.339 |
| FiLM | NL | 1 | 0.746 | 0.620 | 0.691 | 0.691 | 0.707 | 0.392 |
| DAFT Dense | NL | 4 | 0.650 | 0.510 | 0.602 | 0.590 | 0.631 | 0.200 |
| DAFT Dense | NL | 3 | 0.728 | 0.589 | 0.676 | 0.674 | 0.690 | 0.319 |
| DAFT Dense | NL | 2 | 0.747 | 0.590 | 0.688 | 0.694 | 0.694 | 0.360 |
| DAFT | NL | 1 | 0.763 | 0.618 | **0.706** | 0.704 | 0.719 | 0.380 |

to the task which one is trying to optimize. As another dense fusion candidate, Duanmu et al. (2020) performed slightly better, but could not yet outperform image-only ResNet. Given that the tabular data should complement the model and not misguide it, or findings suggested that the late fusion should be favored particularly in cases where the tabular performance is significantly lower than that of the image-only model. However, more experiments on this matter with different relation between image and tabular modalities are needed, e.g., when the tabular features serve as a main decision-driving force, whereas image data provide supplementary information.

## 4. PSA profiles

**Experiments** We investigate the impact of PSA profile on the classification performance of prostate diseases. The longer the PSA history, i.e., the more PSA values available, the smaller the fraction of subjects with a history of that length. This may lead to either a small training data set or a large fraction of missing values. Regarding our data set, we report results for a history of at most three PSA samples, which still gives enough patients for training. To be precise, the fraction of patients with missing PSA samples for one, two, and three PSA samples is 21%, 27%, and 38%, respectively. Since PSA samples are distributed differently over time for each patient, we included the patient's age at the time of PSA collection as indicating feature. This age feature helps to reflect the intra-patient progression of PSA, as well as the inter-patient relation throughout the whole cohort. As a single model will not provide us with a reliable trend in this experiment, we decided to validate it with all models defined previously except FiLM Dense and DAFT Dense.

**Results** In Table 2 we can observe a clear trend that indicates a significant increase in diagnostic performance when using one or two PSA samples. However, with a sample size of three, the performance started to stagnate and no substantial differences were found. We assume that this might be due to a higher fraction of missing samples in longer PSA profiles. Possibly, samples that are not up to date add only little value to diagnostic detection because they do not indicate an actual state of the patient but rather serve to show the progression of PSA. Despite the clear benefits of using PSA information with the fusion method of (Duanmu et al., 2020), it still does not reach the performance of the image-only model. These results also indicate that, when using PSA information, the Concat-1FC and 1FC-Concat-1FC methods slightly outperformed the methods with linear modulation: FiLM and DAFT, but there is no significant difference between Concat-1FC and 1FC-Concat-1FC. However, in this case, the DAFT approach shows superior performance over the FiLM, indicating that image feature modulation derived from two sources is more effective than when only tabular data inform these features.

## 5. Conclusion

Reliable decision making in clinical routines is never possible without taking into account multiple data sources that can differ drastically in their kind of representation. This is especially true for the diagnosis of prostate disease. Our work aims to make one step towards an automatic computer-assisted diagnosis at the patient level by combining MR images, patient demographics, and PSA profiles. As a proof of concept, we compared the baseline

Table 2: Performance on multiclass tasks across five folds on hold-out test set. Column PSAs indicate the number of samples from the PSA profile used for training.

| | | csPCa AUC | | | | | |
| | PSAs | ROC | PR | Precision | Recall | ACC | QWK |
|---|---|---|---|---|---|---|---|
| ResNet | 0 | 0.736 | 0.611 | 0.679 | 0.684 | 0.685 | 0.342 |
| Logistic regression | 0 | 0.651 | 0.505 | 0.607 | 0.584 | 0.636 | 0.227 |
| | 1 | 0.671 | 0.576 | 0.621 | 0.596 | 0.647 | 0.271 |
| | 2 | 0.698 | 0.601 | 0.644 | 0.619 | 0.665 | 0.340 |
| | 3 | **0.704** | **0.616** | **0.648** | **0.620** | **0.668** | **0.339** |
| Concat-1FC | 0 | 0.767 | 0.644 | 0.701 | 0.706 | 0.695 | 0.376 |
| | 1 | 0.780 | 0.656 | 0.719 | 0.720 | 0.730 | 0.394 |
| | 2 | 0.788 | 0.671 | 0.724 | 0.722 | **0.737** | 0.430 |
| | 3 | **0.799** | **0.695** | **0.727** | **0.727** | 0.734 | **0.455** |
| 1FC-Concat-1FC | 0 | 0.773 | 0.665 | 0.711 | 0.706 | 0.721 | 0.419 |
| | 1 | **0.794** | 0.680 | **0.731** | **0.725** | **0.741** | **0.442** |
| | 2 | 0.790 | 0.679 | 0.719 | 0.718 | 0.732 | 0.436 |
| | 3 | 0.790 | **0.681** | 0.721 | 0.722 | 0.730 | 0.432 |
| Duanmu et al. (2020) | 0 | 0.720 | 0.601 | 0.659 | 0.654 | 0.677 | 0.312 |
| | 1 | 0.727 | 0.612 | 0.655 | 0.648 | 0.674 | **0.383** |
| | 2 | 0.699 | 0.573 | 0.634 | 0.622 | 0.657 | 0.328 |
| | 3 | **0.730** | **0.619** | **0.663** | **0.653** | **0.681** | 0.377 |
| FiLM | 0 | 0.746 | 0.620 | 0.691 | 0.691 | 0.707 | 0.392 |
| | 1 | **0.772** | 0.639 | **0.699** | 0.695 | 0.708 | 0.402 |
| | 2 | 0.758 | 0.638 | 0.690 | 0.689 | 0.702 | 0.415 |
| | 3 | **0.772** | **0.646** | 0.698 | **0.698** | **0.709** | **0.458** |
| DAFT | 0 | 0.763 | 0.618 | 0.706 | 0.704 | 0.719 | 0.380 |
| | 1 | 0.789 | 0.666 | 0.716 | 0.711 | 0.725 | 0.462 |
| | 2 | 0.793 | 0.668 | **0.724** | **0.727** | **0.733** | 0.457 |
| | 3 | **0.794** | **0.676** | 0.716 | 0.714 | 0.722 | **0.464** |

model with five recent approaches for the diagnosis of csPCa and the grading of prostate diseases (including prostatitis). We showed that the naïve concatenation approaches outperformed the advanced state-of-the-art modules such as linear feature modulation by a non-significant margin. In addition, the impact of dense fusion on the diagnostic performance was analyzed. We found that dense fusion variants negatively affected the overall performance as it probably put more emphasis on tabular data in our case. Moreover, our research has highlighted the importance of PSA profiles and proposed a simple yet effective way to incorporate PSA changes over time that are not straightforward to summarize. Future work should focus on the impact of using segmentation of the four anatomical zones of the prostate (Meyer et al., 2019) or periprostatic fat (Li et al., 2022) along with mpMRI and their derived volumetric parameters as a tabular feature. Furthermore, the fitness of recent time-series models and how they address missing values should be thoroughly investigated in the context of PSA profiles and multimodal fusion.

## Acknowledgments

This work has been supported, in parts, by the federal state of Saxony-Anhalt (Germany) within the framework of the postgraduates funding and by Central Innovation Programme for small and medium-sized enterprises (SMEs) Germany (16KN093940) in cooperation with ALTA Klink GmbH.

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

## Appendix A. Additional results for experiments on fusion methods (binary task)

Table 3: Performance for binary task averaged across five folds on hold-out test set. Column T marks the linear (L) / non-linear (NL) transformation of tabular data. Column FPs shows the number of fusion points, where FP > 1 means dense fusion.

| | | | csPCa AUC | | | | |
|---|---|---|---|---|---|---|---|
| **Binary** | T | FPs | ROC | PR | Precision | Recall | ACC |
| Logistic regression | L | - | 0.652 | 0.508 | 0.604 | 0.582 | 0.634 |
| ResNet | - | - | 0.736 | 0.611 | 0.663 | 0.647 | 0.650 |
| Concat-1FC | L | 1 | **0.769** | 0.632 | 0.685 | 0.658 | 0.701 |
| 1FC-Concat-1FC | NL | 1 | 0.763 | **0.634** | **0.710** | **0.695** | **0.722** |
| Duanmu et al. (2020) | NL | 4 | 0.724 | 0.576 | 0.651 | 0.629 | 0.672 |
| FiLM Dense | NL | 4 | 0.644 | 0.506 | 0.609 | 0.586 | 0.638 |
| FiLM Dense | NL | 3 | 0.721 | 0.587 | 0.656 | 0.653 | 0.673 |
| FiLM Dense | NL | 2 | 0.726 | 0.592 | 0.663 | 0.641 | 0.673 |
| FiLM | NL | 1 | 0.752 | 0.607 | 0.698 | 0.685 | 0.714 |
| DAFT Dense | NL | 4 | 0.653 | 0.510 | 0.601 | 0.588 | 0.631 |
| DAFT Dense | NL | 3 | 0.731 | 0.589 | 0.677 | 0.675 | 0.691 |
| DAFT Dense | NL | 2 | 0.762 | 0.623 | 0.681 | 0.650 | 0.691 |
| DAFT | NL | 1 | 0.765 | 0.626 | 0.704 | 0.689 | 0.716 |

## Appendix B. Dataset overview

Table 4: Class distribution breakdown.

| Prostatitis | GS6 | GS7a | GS7b | GS8 | GS9+10 | csPCa | Total |
|---|---|---|---|---|---|---|---|
| 1467 | 849 | 809 | 274 | 141 | 260 | 1484 | 3800 |

Table 5: Summary of the demographic and clinical characteristics of the patients.

| Parameter | Median (interquartile range) | # of subjects | % of subjects |
|---|---|---|---|
| Age, $year$ | $66\,(60-71)$ | 3355 | 88 |
| Year of birth, $year$ | $1950\,(1956-1944)$ | 3355 | 88 |
| Height, $m$ | $1.8\,(1.75-1.83)$ | 2245 | 67 |
| Weight, $kg$ | $88\,(80-95)$ | 2245 | 67 |
| BMI, $kg/m^2$ | $27.2\,(24.8-29.32)$ | 2245 | 67 |
| PSA, $ng$ | $6.6\,(4.7-9.5)$ | 3005, 2768, 2357* | 79, 72, 62* |

*Subjects with one, two and three last PSA samples before the MRI acquistion.

## Appendix C. Additional experiments with different backbone

To assess the generalization of the examined fusion methods, we changed the image-based model to the modified ResNet without residual connections (ConvNet). Overall, ConvNet performed slightly worse than ResNet as an image-based baseline and with the fusion methods. However, according to the results of the binary task in Table 6, the ConvNet follows the same trend with respect to the inclusion of tabular data and the number of fusion points as ResNet. Significant improvement was achieved by adding tabular information to the model. Dense fusion models with 3 and 4 fusion points could not surpass the performance of the image-only baseline model. Generally, as a fusion method, DAFT performed better than FiLM and showed promising results even with two fusion points. Topologically different NN models, such as the Vision Transformer (ViT), will be examined in future work.

Table 6: Performance for binary task averaged across five folds on hold-out test set using the ConvNet model. Column T marks the linear (L) / non-linear (NL) transformation of tabular data. Column FPs shows the number of fusion points, where FPs > 1 means dense fusion.

| | | | csPCa AUC | | | | |
| **Binary** | T | FPs | ROC | PR | Precision | Recall | ACC |
|---|---|---|---|---|---|---|---|
| Logistic regression | L | - | 0.652 | 0.508 | 0.604 | 0.582 | 0.634 |
| ConvNet | - | - | 0.727 | 0.585 | 0.656 | 0.617 | 0.670 |
| 1FC-Concat-1FC | NL | 1 | 0.759 | **0.630** | 0.679 | 0.654 | 0.675 |
| Duanmu et al. (2020) | NL | 4 | 0.741 | 0.611 | 0.673 | 0.656 | 0.686 |
| FiLM Dense | NL | 4 | 0.642 | 0.505 | 0.607 | 0.590 | 0.636 |
| FiLM Dense | NL | 3 | 0.705 | 0.565 | 0.625 | 0.622 | 0.646 |
| FiLM Dense | NL | 2 | 0.717 | 0.589 | 0.656 | 0.647 | 0.672 |
| FiLM | NL | 1 | 0.752 | 0.626 | **0.706** | 0.622 | 0.676 |
| DAFT Dense | NL | 4 | 0.706 | 0.569 | 0.652 | 0.642 | 0.670 |
| DAFT Dense | NL | 3 | 0.716 | 0.589 | 0.662 | 0.659 | 0.678 |
| DAFT Dense | NL | 2 | 0.747 | 0.605 | 0.685 | **0.688** | **0.696** |
| DAFT | NL | 1 | **0.772** | 0.626 | 0.683 | 0.655 | 0.680 |

## Appendix D. Qualitative visualization with Grad-CAM

Grad-CAM (Selvaraju et al., 2017) heat maps were used to visualize the important features that contributed to the prediction for the binary task. As Figure 2 shows, cases a) and b) benefited from tabular information where the image-based ResNet model misclassified it. Subject c) was correctly classified by all models; however, the Grad-CAM emphasized features more on the inside of the prostate gland than on the outer structures. Case d) demonstrated that only FiLM and DAFT were correctly classified, while all heatmaps highlight the same region. However, the tabular fusion misclassified case e), while the image model predicted the correct class. For patient f), all models failed to predict the correct class, even though the Grad-CAM could emphasize the relevant features more clearly.

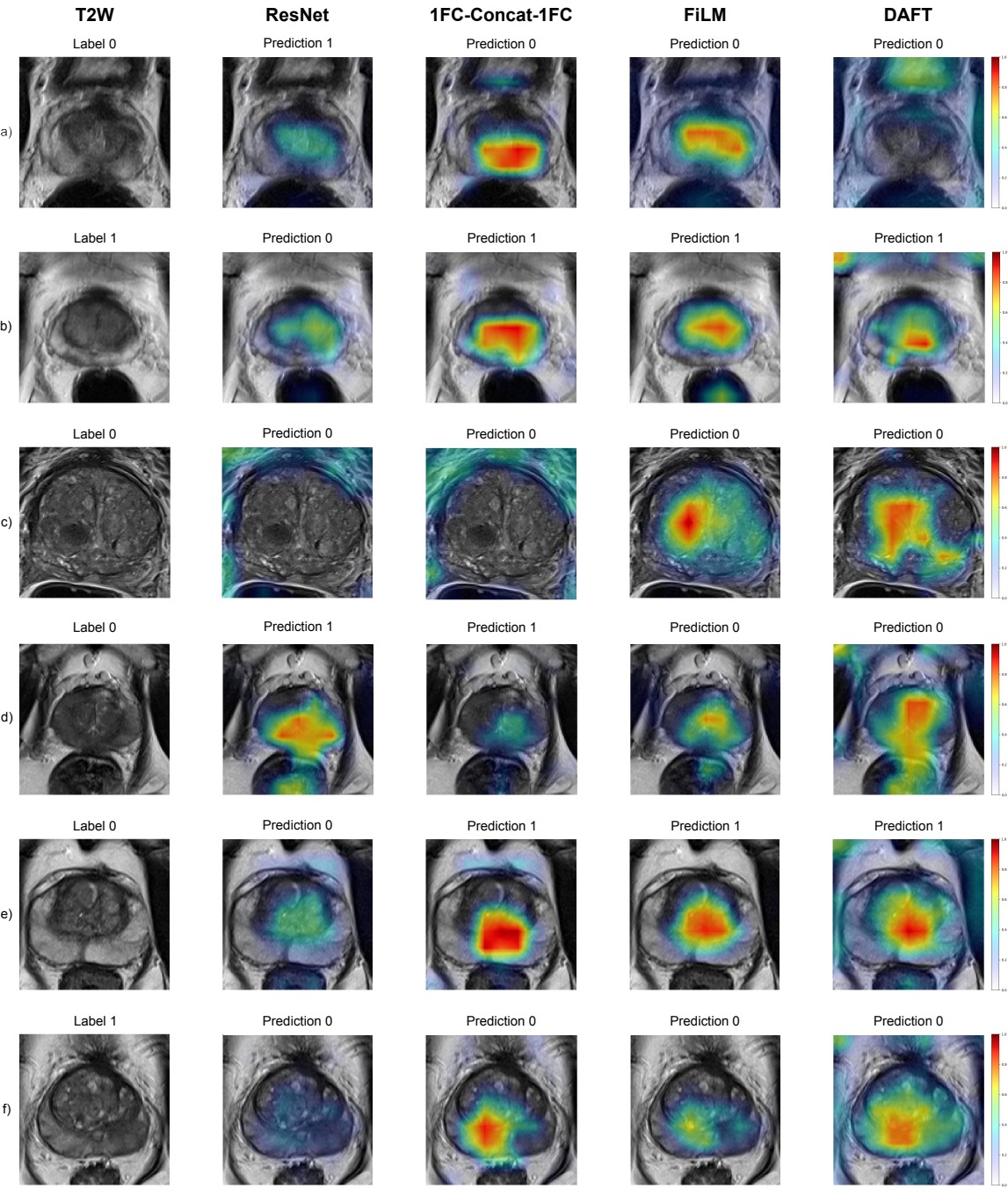

Figure 2: Qualitative visualization of the learned features for the ResNet, 1FC-Concat-1FC, FiLM, DAFT models with the Grad-CAM heat maps.

