# OpenReview forum: "Automatic Patient-level Diagnosis of Prostate Disease with Fused 3D MRI and Tabular Clinical Data"
_MIDL.io/2023/Conference — MIDL 2023 Poster_

### Official Review · Reviewer_o74H · 2023-02-03

**Confidence:** 3
**Preliminary Rating:** 4
**Recommendation:** Poster

**Summary:**

The topic of fusion of heterogeneous data sources in medical imaging applications is a very important one. This paper evaluates several methods for fusing MRI and tabular data for the application of prostate cancer diagnosis. Interestingly, the baseline methods of fusion (based on feature concatenation) work the best. I find the topic of the paper to be very relevant.

**Strengths:**

- Good experimental setup.
- Two models in addition to baselines are investigated. Furthermore, several variants of the models (different concatenation points) are also investigated.
- The authors also investigate the added value of PSA profiles.
- Relatively large dataset.

**Weaknesses:**

I have to remarks regarding the presentation of the methods and results (these are not major weaknesses, however):
- I would add a figure with overview of the methodology. This will significantly improve the clarity and readability of the paper (even if it is added as an appendix, due to page limit).
- The paper tries to tell two stories at the same time: investigating the best way for combining tabular and imaging data and investigation the added value of the PSA profiles. This lack of focus makes it a bit difficult to read.

**Deanonymize Review:**

no

**Paper Type:**

validation/application paper

**Questions To Address In The Rebuttal:**

Addressing the two points mentioned above will improve the clarity of the paper:
- Adding an overview figure of the methodology.
- Focusing the paper on the investigation of the different methods for model fusion.

---

### Official Review · Reviewer_Ew2G · 2023-02-04

**Confidence:** 5
**Preliminary Rating:** 4
**Recommendation:** Poster

**Summary:**

This paper presents a diagnostic model for the classification of prostate cancer lesions and characterization of their aggressiveness (Gleason score) fusing information provided by MR T2w and ADC images as well as clinical data including the standard Prostate-specific antigen (PSA) marker. The backbone architecture is that of a 3D ResNet. The tabular information are encoded by different methods and injected at different levels of the backbone CNN (dense, late fusion). Impact of these  different methods is assessed through an experimental validation study including 3800 private MRI studies. Results indicate performance gain achieved by the fusion of the clinical and imaging data, especially when including PSA. The best reported performance is achieved with simple strategies of clinical feature concatenation.

**Strengths:**

-The paper addresses a hot topic of the community concerning the fusion of heterogeneous modalities (image and clinical data) into deep architectures

-The experimental validation is well-conducted on a rich private database encompassing 3800 MR exams with corresponding Gleason score as well as demographic and PSA data.


**Weaknesses:**

-The paper does not contain strong novel methodological contribution, but I guess this is balanced by the quality of the experimental study.

-The paper lacks description of the fusion strategy as well as methodological details (See specific comments below)

-I could not find any discussion element on the way the model handles patient exams containing multiple lesions of different Gleason score.


**Deanonymize Review:**

no

**Detailed Comments:**

-The paper compares state-of-the art fusion models such as FILM and DAFT, which the community may not be familiar with. I guess unexpert readers would greatly appreciate getting an idea of the basics of such methods without reading the full seminal paper. I suggest the authors to add some illustrations in the Appendix section.

-Please clarify how you computed ROC AUC and PR AUC for the multiclass task. The sentence
‘We furthermore mapped the multiclass task into binary csPCa classification by treating the probabilities of Gleason scores of GS7a and above as csPCa’ is not clear. In the multiclass setting, I assume the model will output a vector of dimension equals to the number of Gleason class, where each value correspond to the softmax probability. Please clarify how these multiclass probabilities are converted into binary prediction probabilities, by summing the probabilities of Gleason score higher than GS7a or by considering the Gleason score with the maximum probability?

-Two questions regarding how the PSA values are injected into the model. First, the authors mention in section 2.2 that ‘”they included the patient’s age at the time of PSA collection as indicating feature”.  How was this implemented? Did the authors add a 6-length dimension vector to the 10-length demographic data, encoding for the PSA values and related patient’s age?  Second, it is not clear how missing PSA values are estimated, since these values depend on the patient’s age at collection time. Please clarify.



**Paper Type:**

both

**Questions To Address In The Rebuttal:**

Please discuss the points i listed as 'weak' , regarding description of the fusion methods as well as concerning the analysis of exams containing multiple lesions. Also please address questions of the 'comment' section.

---

### Official Review · Reviewer_hsTu · 2023-02-07

**Confidence:** 5
**Preliminary Rating:** 2

**Summary:**

The paper introduced a feature fusion method on 3D MRI and clinical data for large-scale prostate disease classification tasks. The clinical data is in a tabular format, including combinations of three PSA profiles and regular demographic data. The validation focus on both binary and multi-class tasks with sensitivity analysis on the usage of PSA features.

**Strengths:**

The author introduced the clinical background of the work and related work, such as the prostate-specific antigen (PSA) test and the Glean Score (GS). The pre-processing step for the MR data is clear and well-written.

**Weaknesses:**

The readability of the manuscript is very poor. There are no figures (and, of course, none of the medical imaging figures are shown) for reference make. A few clarity pr need to be solved (please check questions to address in the rebuttal session).

**Deanonymize Review:**

no

**Detailed Comments:**

To improve the readability of the work, authors could add a few figures (at least method figure, experiment design and qualitative figures).

**Paper Type:**

validation/application paper

**Questions To Address In The Rebuttal:**

- 640 cases were left as a hold-out test set. Why authors only showed 5-fold cross-validation results (which is totally fine), but without test results?

- The Restnet is a great framework for classification tasks. However, since the 'meat' of the paper is to utilize tabular data, it is unclear if the tabular features can reproduce similar performance on other networks.

- 'Thus, we have 10 features in total' - this is a minor notice; could authors clarify how many features exactly are? It seems to be 8: (1) age, (2) birth year, (3) height, (4) weight, (5) BMI, (6-8): PSA.

---

### Meta-Review · Area_Chair_cjPP · 2023-02-24

**Recommendation:** Accept (Poster)
**Confidence:** 4

**Metareview:**

Even though the reviewers initially had some moderate concerns, the authors adequately addressed their comments. The reviewers highlight the importance of the topic tackled, the fusion of heterogeneous data, and the quality of the experimental study.